# Competence Development and Collaborative Climate as Antecedents of Job Performance, Job Commitment and Uncertainty: Validation of a Theoretical Model across Four Hospitals

**DOI:** 10.3390/ijerph20010425

**Published:** 2022-12-27

**Authors:** Maria Therese Jensen, Olaug Øygarden, Aslaug Mikkelsen, Espen Olsen

**Affiliations:** 1Centre for Learning Environment, University of Stavanger, 4036 Stavanger, Norway; 2NORCE, 4021 Stavanger, Norway; 3Department of Innovation, Leadership, and Marketing, UiS Business School, University of Stavanger, 4036 Stavanger, Norway; 4Stavanger University Hospital, 4011 Stavanger, Norway

**Keywords:** human resource management, competence development, organizational climate, job commitment, job performance, uncertainty, hospital

## Abstract

Knowledge is lacking regarding how organizational factors are associated with uncertainty in patient treatment. Thus, the aim of the current study was to investigate how competence development and collaborative climate relates to job performance and job commitment, and further whether job performance and job commitment relate to uncertainty. Additionally, we examined whether these associations differed between four different hospitals. We applied data from 6445 hospital workers who provided care to patients. Basic statistics and structural equation modelling (SEM) were used to test the validity of the theoretical model developed in the study and the hypothesized associations. All hypothesized paths between the latent variables were significant and in accordance with the model across the four hospitals. The current study has implications for practical human resource management and indicates that competence development should be strengthened at the individual level and collaborative climate should be strengthened at the ward level. Strengthening competence development and collaborative climate can increase job performance and job commitment of individual workers and reduce uncertainty during care in hospital settings.

## 1. Introduction

The quality of health care systems is determined by a range of factors including funding principles, organizational models, medical technical equipment, and communication systems. Quality in health care is, however, ultimately dependent on the health care professionals who deliver care. Their performance and commitment to the job is decisive for the quality of the service provided. A better understanding of how organizational factors such as organizational climate and human resource management (HRM) practices influence health service professionals’ performance, commitment and uncertainty is, therefore, crucial to the quality of health care systems and to our understanding of work conditions in hospital settings. 

The problematic nature of uncertainty in health care has been acknowledged by scholars since the 1950s. Sociologist Rene’e Fox [1] demonstrated in his studies how physicians struggle with uncertainty during their training. Recently, uncertainty in health care has attracted growing attention [2]. “Uncertainty is an essential aspect of human life and an integral problem of medicine. It is the single, common challenge faced by every patient who receives health care and every clinician who provides it, as well as the administrators, payers, policymakers, and researchers who deliver, finance, regulate, and study it. In every one of these diverse activities undertaken by different stakeholders, uncertainty of one form or another—arising from various sources, pertaining to any number of relevant issues, and formed and reformed through communication” [3] (p. 1756). A more systematic research agenda has been called on in relation to uncertainty in health care. Moreover, “uncertainty in health care is still an extremely important but incompletely understood phenomenon”, and improving our understanding of the many important aspects of uncertainty in health care is crucial towards improving care [3] (p. 1756). Hence, the aim of the present study was to develop and test a theoretical model of organizational antecedents of uncertainty among staff in hospital settings, and to test this model across four hospitals. Specifically, we investigate how competence development and collaborative climate relates to job performance and job commitment, and further whether job performance and job commitment relates to uncertainty. We split assessments across four hospitals to potentially cross-validate the theoretical model developed and increase the generalizability and robustness of our research findings.

## 2. Theoretical Background and Hypothesis Development

### 2.1. Uncertainty in Patient Treatment and Its Determinants

Task uncertainty can be defined as being the difference between the amount of information required to perform a task and the amount of information already possessed by the organization [4]. Due to the complexity in healthcare organizations, workers are exposed to high levels of uncertainty at work [5]. Uncertainty can have negative impacts on individual’s behaviors, as it can cause feelings as anxiety, fear and further avoidance in making decisions [5]. Uncertainty may be determined by a number of determinants. Most determinants work through human resource management (HRM). HRM include human resource practices that are designed to improve individual performance and, through this, organizational performance [6]. The main conclusion when looking at over 300 published articles on HR strategy since the early 1990s is that the financial returns of investment in high-performance work systems are both financially and statistically significant [7]. The evidence that high performance HRM practices evokes positive outcomes has led the focus towards trying to explain why these systems evoke positive outcomes [8,9]. Numerous theoretical frameworks have been applied to explain this relationship, for example social exchange theory [10], the norm of reciprocity [11], ability–motivation-opportunity (AMO) theory [12], self-determination theory [13] and affective events theory [14]. HRM at the individual level involves managerial attempts to influence employee abilities and the knowledge, skills and attitudes required for problem solving, carrying out job tasks and achieving the goals of the organization. It also involves motivating employees by setting goals, giving feedback and creating opportunities to participate, which in a hospital setting means collaboration with others in achieving the joint-held goal of improving the health of patients. This line of argument is in line with AMO theory [12]. To prevent health workers from experiencing uncertainty in patient treatment, it is crucial that HRM is continuously developed.

### 2.2. Collaborative Climate as an Aspect of Organizational Climate

Organizational climate has been defined as “the enduring quality of an organization’s internal environment, distinguishing it from other organizations, which results from the behavior and policies of members of organizations” [15] (p. 126). Organizational climate is a key explanatory variable in understanding the effect of HRM practices on individual outcomes [16]. Climate can also be seen as an intervening variable, which may be affected by a set of external and structural variables and simultaneously has an influence on organizational outcomes such as employee-related performance and behavior [17]. People in organizations encounter thousands of events, practices, and procedures, and bring sense to this diversity by perceiving these events in related sets. This implies that a work setting can have different organizational climates and that each climate serves a different purpose [18]. Thus, organizational climate can be considered a multidimensional construct. For instance, in a work setting there may be a general work climate [19], one climate for service [20,21], a climate for safety [22,23,24,25,26,27], another climate for collaboration [28], and so on. According to this line of thought, it makes sense to define collaboration climate as an aspect of organizational climate. 

A presumption in traditional leadership research is that leadership, by definition, is performed by a formal leader who influences or transforms members of a group or organizations to achieve specified goals [29]. This presumption does not automatically apply in a collaborative setting. This is because the individuals involved come from different parts of the organization, from different professional groups or from other organizations. Collaborative advantage has been introduced as an organizational phenomena, which indicates that certain achievements could not have been made without the collaboration [30]. Collaboration can be considered a form of social capital that supports collective action and change through dense networks of information sharing, trust and reciprocation norms [31]. Summarized, a strong collaborative climate is associated with many positive organizational aspects that influence individual workers.

### 2.3. The Importance of Competence Development and Its Relation to Collaborative Climate

The main aim of HRM is to secure the competence required for the organization to deliver the services it is expected to provide. However, changes in medical treatment, equipment and new expectations of the population implies that health services are dependent on continuous competence development. Competence is a key factor in motivation theories [12,13,32]. There are certain basic needs that must be satisfied for individuals to obtain intrinsic motivation, where competence is one of these [13,32]. Specifically, competence relates to the need for a sense of proficiency and feelings of effectiveness in one’s work [32] and is additionally related to personal growth and the challenging of abilities and knowledge [33]. Employees who work in an environment where they can perform optimally and enjoy their work, will be highly motivated and function well as employees. A focus on learning and competence development therefore represents the main competitive advantage of the quality enhancement strategy [34] followed by many hospitals. Competence development plans are a key tool for establishing learning goals and for employees to maintain and develop the capabilities of human resources. Moreover, positive emotions can be evoked when an employer helps employees to meet their need for competence development. According to the broaden-and-build theory [35] positive emotions widen the arrays of thoughts and actions, and ensures personal resources such as social connections, coping strategies and environmental knowledge. Prioritizing competence development in a hospital setting means theory input, practical on the job training and experience with co-workers in teams. Accordingly, competence development has the potential to improve the collaborative climate among staff. Thus, we propose the following hypothesis: 

**Hypothesis** **1:**
*Competence development relates positively to collaborative climate.*


### 2.4. Competence Development and Collaboration Climate as Antecedents of Job Performance and Job Commitment and the Implications for Uncertainty

Job commitment expresses the strength of the employees’ emotional attachment to an organization and acceptance of the organization’s goals and values [36]. This emotional attachment is, however, not without costs and the relation has an element of exchange [37]. Employees come to organizations with certain needs and resources, and therefore expect to find a work environment in which they can utilize their abilities and have their needs satisfied. Self-determination theory has been applied to integrate commitment into work motivation theories [38]. Generally, employees are more likely to invest the resources required to initiate or maintain commitment when their needs are met. Competence development and a collaborative work climate satisfy certain psychological needs and provide a work environment in which employees can use their abilities [32]. Accordingly, we propose the following hypotheses: 

**Hypothesis** **2:**
*Competence development is positively associated with job commitment.*


**Hypothesis** **3:**
*Collaboration climate is positively associated with job commitment.*


Self-evaluation of job performance can be considered an affective process, and persons behave in a manner that will maintain or increase self-evaluation. Positive affect describes an individual’s tendency to be energetic and experience positive moods and emotions across different situations. Competence development may strengthen self-esteem and lead to positive affect. Positive affect also influences critical organizational outcomes such as performance [39], and we therefore expect to see a positive association between competence development and job performance. Organizational climate of collaboration includes knowledge sharing [40,41], pro-social behavior [42] and higher levels of giving in a work situation [43,44]. This implies that members of a group are more inclined to work together, to share and develop tacit knowledge and to promote each other’s performance and learning [45]. However, it is reasons to believe that collaboration climate also will have a positive association with job performance at the individual level. A recent study demonstrated that interdepartmental collaboration climate was positively associated with organizational attractiveness and service quality of care [46]. Hence, based on the theories presented above, we suggest the following hypotheses: 

**Hypothesis** **4:**
*Competence development is positively related to job performance.*


**Hypothesis** **5:**
*Collaboration climate is positively related to job performance.*


Finally, we also expect to see that job performance and job commitment relates negatively to uncertainty. It has been suggested that factors related to the work environment has a significant impact on patient care (Cummings [47] et al., 2010). Moreover, it has been demonstrated that satisfying psychological needs are negatively related to uncertainty among nurses and physicians [5]. Job commitment and job performance reflects positive emotional states among workers, e.g., [48,49] and therefore we expect these positively connotated job concepts to reduce uncertainty in hospital setting. Thus, we propose the following hypotheses: 

**Hypothesis** **6:**
*Job commitment is negatively related to uncertainty.*


**Hypothesis** **7:**
*Job performance is negatively related to uncertainty.*


Finally, demographic factors may influence the dynamics of organizations. Gender and number of working hours were therefore included as control variables in the estimations. Figure 1 shows the theoretical model of our study, and illustrates the hypotheses presented above. 

## 3. Materials and Methods

### 3.1. Sample and data Collection

We collected data from four public hospitals in a Norwegian regional health authority. The regional health authority has more than 20,000 employees and provides services to 1.1 million citizens. The study was conducted in October 2014 and data were collected via an internal web-application distributed to all health care employees of the health authority. All survey responses from the informants were anonymous. The survey consisted of a range of validated questions on themes relevant to the two issues. The overall response rate was 40 percent (N = 9162). A total of 6445 employees were included in the study based on the criterion that those included should provide patient treatment and care. Only personnel responding “yes” to the item “Do you have direct patient contact” in the survey were included in the study. Service and administrative personnel were therefore not included. The respondents were distributed as follows across the four hospitals: hospital 1 (N = 2824), hospital 2 (N = 822), hospital 3 (N = 1134), hospital 4 (N = 1655). Of respondents, 83.1% were female, 56.1% were more than 40 years of age, 8.6% were physicians, 45.7% were nurses, and 30.8% worked 4 h or less a day. 

### 3.2. Measures

Four items from the COPSOQ instrument were applied to measure competence development [50,51]. The items assess the opportunity to learn, adapt and use work-related skills and expertise. Items are measured on a 5-point scale ranging from strongly disagree to strongly agree. 

Five items were applied to measure collaborative climate [51,52]. Statements and descriptions used to measure collaborative climate included whether the climate is characterized by (1) coldness and condescension, (2) lack of trust, (3) complaining behaviors, (4) inter-professional collaboration, and (5) warmth and positivity among co-workers and leaders. Three of the items were reversed before assessment because of negative statements and semantics in the items. Items are measured on a 5-point scale ranging from strongly disagree to strongly agree. Three items were used to measure job commitment [53]. The items measure job commitment in terms of a feeling of belonging to the workplace, personal value in the job and a feeling of pride in the job. Items are measured on a five-point scale ranging from very low degree to very high degree. Four items from QPS Nordic were used to measure job performance [54]. The scale uses an employee self-assessment of their job performance. Topics include quantity of work, quality of work, the ability to solve problems at work and satisfaction with one’s capacity to develop and maintain good work relationships with colleagues. Items were measured on a five-point scale ranging from never/seldom to always/very often. Uncertainty in patient treatment was measured using five items from the Nursing Stress Scale (NSS) [55] and an item on lack of personnel. Item scales ranged from 1 (never) to 4 (very often). They are preceded by the statement; “How often do you experience the following situations” and followed by the item, such as “Prescribing the wrong treatment of patients”. Uncertainty items also covered challenges relating to insufficient information, wrong treatment, non-available physicians, staff shortages and providing patients with the correct information.

### 3.3. Data Analysis 

The data were analyzed using version 21.0 of SPSS and AMOS 21.0 [56]. SPSS was used to analyze basic descriptive statistics, bivariate correlations and Cronbach’s alpha. AMOS was used to carry out confirmatory factor analyses (CFA) and structural equation modelling (SEM). Correlations between concepts indicate level of difference and discriminant validity of concepts.

## 4. Results

### 4.1. Descriptive Statistics, Correlations and Internal Consistency

Descriptive statistics for the variables for the four hospitals are presented in Table 1. The mean score for the variables ranged from 1.14 to 6.50. The standard deviations for all items were satisfactory. 

Table 2 shows Cronbach’s alpha and correlations for the total sample. Alpha scores ranged from 0.74 to 0.89 and were considered to be satisfactory. Correlations ranged from −0.20 to 0.33 (*p* < 0.01, two-tailed). Both the reliability and correlation estimates were, in general, considered to be adequate. 

### 4.2. Confirmatory Factor Analysis and Structural Relationships 

CFA indicated a satisfactory fit between the measurement concepts in Figure 1 and the data across the total sample (N = 6445); CFI = 0.93, IFI = 0.93, TLI = 0.91, NFI = 0.93, RMSEA = 0.043 (90 % confidence interval = 0.042–0.045), Chi-square = 3073.861, degrees of freedom = 233, probability level = 0.000). 

In the next step, we tested the structural associations between measurement concepts, as formulated in our hypotheses, using structural equation modelling (SEM). Results demonstrated all hypothesized paths between the latent variables were significant and in accordance with the proposed hypotheses across all the included hospitals. More specifically, this implies that competence development was positively related to collaborative climate, job performance and job commitment, whereas collaborative climate was positively associated with job commitment and job performance. The two control variables, gender and working hours, had some significant relations with competence development and collaboration climate (Figure 2); in hospital 1 and 2, whereas number of working hours were negative related with and collaborative climate. Further, in hospital 1 and 4, gender was negatively related with collaborative climate, implying that men, perceive the competence development to be lower in the first and forth hospital, compared to women. Further, according to expectations both job performance and job commitment were negatively associated with uncertainty. The explanatory power of the model estimations was relatively similar across all four hospitals and the models had acceptable model fit. The relationships between the latent factors in the model with significant also with control variables (number of working hours and gender).

## 5. Discussion

In the current study we investigated how competence development and collaborative climate relates to job performance and job commitment, and further the significance of job performance and job commitment for employees’ experience of uncertainty during patient treatment. Additionally, we examined whether these associations differed between four different hospitals. In line with expectations, we found support for our hypothesized associations across all four hospitals. This study confirms that individual and organizational level variables are closely related. Specifically, levels of competence development at the individual level, were positively related to collaboration climate at the organizational level. Moreover, both competence development and collaboration climate were positively related to job performance and job commitment. Finally, uncertainty was measured at the individual level, but can be considered an organizational outcome at the aggregated level. An essential aim for health service and health organizations should be that their staff are confident and certain about the services and care they provide. Hence, this study makes important contributions related to an increased understanding with regard to individual and organizational factors influencing the quality of health care services and levels of uncertainty among staff providing care. 

First, and in line with expectations, our findings showed a positive association between competence development and collaboration climate. In line with the broaden and build theory [35] one explanation for our finding is that competence development evokes positive emotions, which again leads to improved social connections among hospital workers. Further, our results revealed positive relations between competence development and job commitment, and collaborative climate and job commitment. This finding is in line with self-determination theory [13,32] and further that employees feel more committed to the organization when their needs are met through for instance competence development and high collaboration climate [38]. Moreover, previous studies have shown that HRM practices such as job enrichment increase employee job commitment [57]. Competence development and collaboration climate can be considered aspects of job enrichment, which can explain the findings in our study. 

Moreover, competence development and collaboration climate were also found to associate positively with job performance. Previous research suggests that positive emotions and affect can have a positive impact on job performance [39]. When employees perceive to have high degree of competence development and a sound collaborative climate, positive emotions may be evoked, which can explain the positive association found between competence development and job performance, and collaborative climate and job performance in the current study. Moreover, investment in employee development facilitates greater employee obligation to the organization and therefore increases the employees’ motivation to work hard and to work for increased organizational effectiveness [58], which again may have positive spillover effects for job performance. 

Learning and competence development are large research topics involving multiple disciplines. Organizational learning theory can be considered a key factor to improve our knowledge regarding competence development. Drejer [59] defines competences as a system of technology, human beings, organizational (formal) and cultural (informal) elements, as well as the interactions of these elements. According to Drejer [59], human beings are the focus when it comes to competence and competence development, and claims that if humans do not apply the technologies, then changes will not happen. There can also be great variation on the combinations of human resources and technology, which is why a systems perspective is needed when considering competence [60]. For instance, a system can consist of a single technology with a few people, interwoven technologies in a larger organizational unit, or complex systems connecting many persons in different departments and organizational units [61]. Organizational learning is typically divided into individual, group and organizational learning [59]. Hence, efforts to strategically increase competence development into hospitals needs to take these levels into consideration, as well as the inclusion of other system components mentioned above which might be related to competence. Moreover, interventions might therefore target multiple interrelated factors which potentially have reciprocal effects. An illustration of this example can be when a hospital decides to couple up junior and senior staff to increase the competence development of junior staff. This situation will force staff to collaborate, which again might improve both competence development and collaboration climate. Hence, this example might also illustrate why competence development is related to other aspects of the social system, as for instance collaboration climate.

Finally, result showed that job performance and job commitment were negatively related to uncertainty. This finding is partly in line with a recent study where psychological needs related negatively to uncertainty among nurses and physicians [5]. However, to our knowledge, previous research has not yet demonstrated that uncertainty is linked to job performance and job commitment in particular. Hence, our finding contributes to the research field and demonstrates theoretical links and the nomological network of uncertainty in health care settings. One explanation for the negative association found between job performance and uncertainty could be that the measure of job performance reflects both control and mastery of work tasks, including experience and competence, which obviously will reduce the levels of uncertainty involving challenging tasks. Similarly, to job performance, job commitment has also several positive connotations, and employees who are committed are also more integrated and connected at work, which might again explain its association with uncertainty.

### Limitations and Future Research

One limitation of our research is the use of cross-sectional data to investigate relationships among study concepts. Therefore, evidence related to causality is limited [62]. Future research should consider a longitudinal study design and explore the long-term effects of competence development initiatives, as well as interventions improving collaboration climate, and explore different strategies to improve levels of job performance, job commitment and uncertainty.

The second limitation of this study it the study setting, hospitals in Norway. Forthcoming studies should replicate this study in other cultures and settings, to explore the generalizability of findings. This will generate knowledge on the potential generalizability of study findings.

A third limitations relates to included study variables; future studies should investigate how uncertainty relates to other factors not included in this study, such as for instance psychological need satisfaction [5], human capital [63], supportive leadership [64], ethical dilemmas and institutional stress [65], job stress [66], safety culture [23,25,26,67], procedures and compliance [68], formal systems [69], work climate [19], job satisfaction [19], work engagement [70], bullying [71], use of technology [72], patient experiences [21], changes processes [73,74] and improvement initiatives [75,76].

## 6. Conclusions

The present study contributes to the understanding of the relationship between aspects of HRM and uncertainty in hospital settings. To our knowledge, this is the first study exploring relations between competence development, collaboration climate, job performance, job commitment and uncertainty in a health care setting and in general. It is interesting to see that the factors included in our study has substantial power in explaining the variance in uncertainty among hospital staff. One strength of the study relates to the testing of our theoretical model across four hospitals that are governed by the same laws and regulations but are managed by different line managers. Conclusively, the current study contributes to understanding the nomological network related to uncertainty and allows for the generalizability of the theoretical model within a Norwegian and possibly Scandinavian context. However, our findings may have both policy and practical implications also outside the Scandinavian context.

## Figures and Tables

**Figure 1 ijerph-20-00425-f001:**
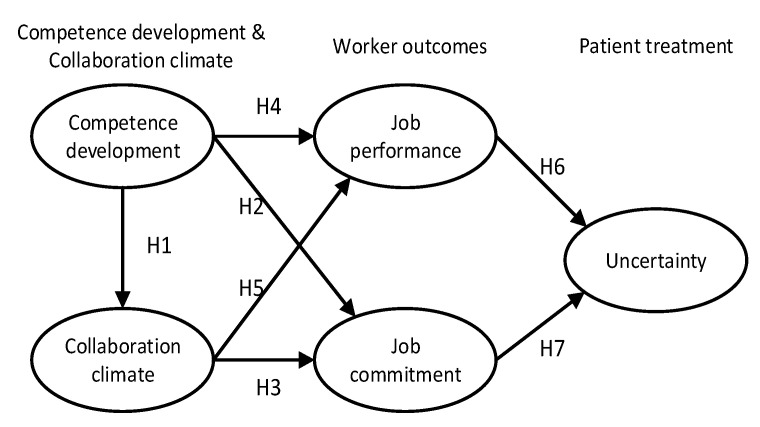
The theoretical model underlying the study.

**Figure 2 ijerph-20-00425-f002:**
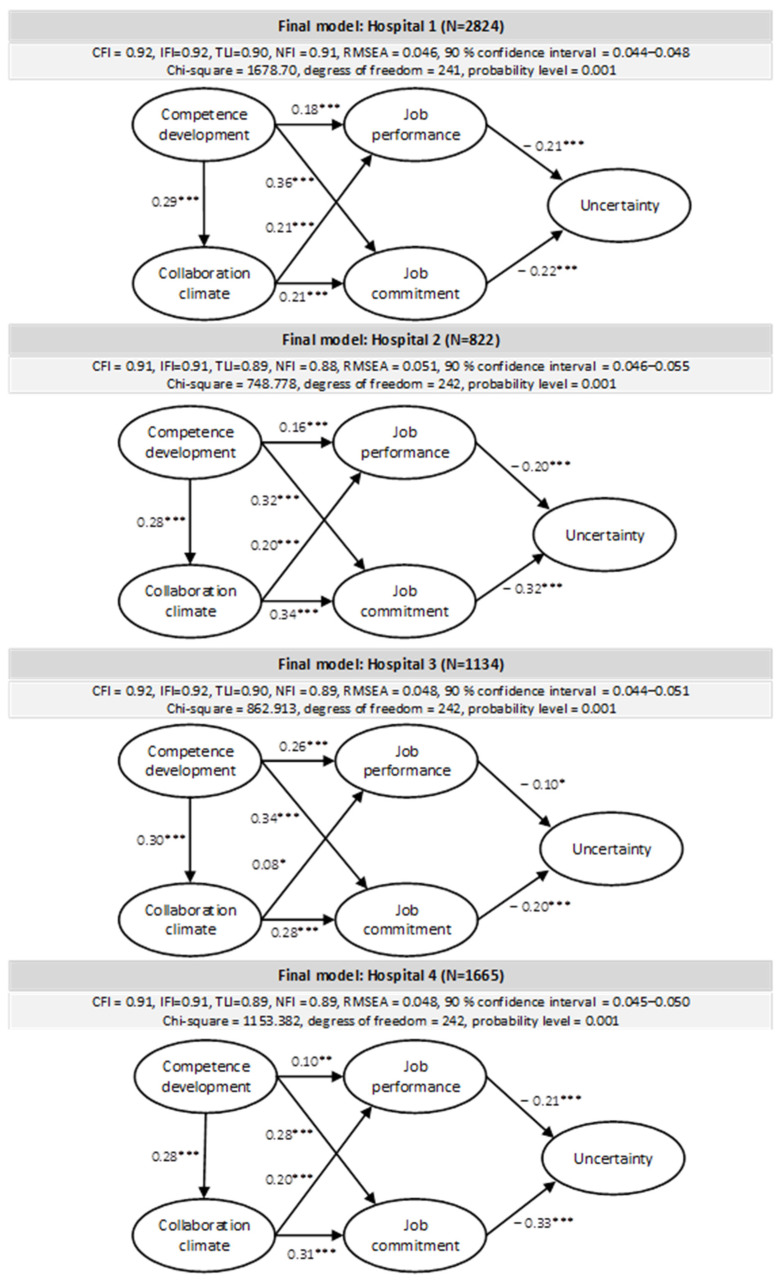
Structural modelling assessed on four different hospitals. Notes; *** *p* < 0.001, ** *p* < 0.01, * *p* < 0.05).

**Table 1 ijerph-20-00425-t001:** Descriptive statistics.

	Hospital 1		Hospital 2		Hospital 3		Hospital 4		Total	
	Mean	SD	Mean	SD	Mean	SD	Mean	SD	Mean	SD
Gender	1.18	0.39	1.19	0.4	1.15	0.36	1.14	0.35	1.17	0.38
Number of working hours	33.95	9.09	33.79	10.70	32.45	8.93	33.11	9.74	33.47	9.45
Competence development	4.37	0.58	4.36	0.60	4.35	0.59	4.37	0.60	4.36	0.59
Collaborative climate	3.47	0.70	3.35	0.73	3.42	0.70	3.45	0.70	3.44	0.71
Job performance	4.11	0.48	4.07	0.50	4.05	0.48	4.10	0.46	4.09	0.48
Job commitment	3.89	0.91	3.84	0.95	3.78	0.97	3.83	0.92	3.85	0.93
Uncertainty	1.79	0.43	1.81	0.43	1.84	0.42	1.87	0.44	1.82	0.43

Notes; Gender: woman = 1, man = 2.

**Table 2 ijerph-20-00425-t002:** Correlations and Cronbach’s alpha (diagonal).

	1	2	3	4	5	6	7
1. Gender	-						
2. Number of working hours	0.05 **	-					
3. Competence development	–0.06 **	–0.02	(0.77)				
4. Collaborative climate	0.05 **	0.01	0.30 **	(0.74)			
5. Job performance	0.00	–0.01	0.17 **	0.26 **	(0.78)		
6. Job commitment	–0.03 *	0.00	0.33 **	0.24 **	0.27 **	(0.89)	
7. Uncertainty	0.03 *	0.06 **	–0.09 **	–0.08 **	–0.15 **	–0.20 **	(0.89)

Notes. Gender: woman = 1, man = 2. * *p* < 0.05, ** *p* < 0.01.

## Data Availability

Not applicable or agreed with the Western Norway Regional Health Authority.

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
