# Peer review of "Competence Development and Collaborative Climate as Antecedents of Job Performance, Job Commitment and Uncertainty: Validation of a Theoretical Model across Four Hospitals"

_ijerph, 2022, doi:10.3390/ijerph20010425_

Round 1

Reviewer 1 Report

The presented work is part of the search for factors determining the effectiveness of high-performance systems. An attempt was made to determine the impact of commitment and uncertainty, through organizational factors such as organizational climate and human resource management (HRM) practices, on the performance of health professionals. According to the authors, these factors are of fundamental importance for the quality of health care and working conditions in closed health care.

The authors made a substantively correct presentation of the literature, including the conclusions resulting from the reviews of the theory of motivation. The examined categories were defined unequivocally and transparently. The research model was clearly defined, and its research elements were created on the basis of qualitative analysis (questionnaires with ratings on a 5-point Likert scale). The method of structural equation modeling (SEM) is adequate to the adopted research methodology. Conclusions in line with the scope of the research

The research results show two trends known from various studies. The first is to increase employee engagement through some HRM practices, such as job enrichment. Secondly, a negative relationship between work efficiency and professional involvement with uncertainty was confirmed. Increased professional experience and competences can reduce this phenomenon, especially in the healthcare sector.

The presented work is one of the first to consider the researched relationships in a specific and important area of ​​human life, which is health care. Moreover, in a specific, broad sense. The work therefore fills the literature gap, contributing to the development of knowledge. For these reasons, I recommend its publication

Author Response

Response: Thank you very much for the positive feedback on this manuscript. We have done some changes to improve the paper even further.

Reviewer 2 Report

It seems to me that the organizational approach in the care of patients in hospitals is novel, since it is of great theoretical and practical relevance.

Breadth in the number of participants

The temporality of the study generates doubt and concern in me, since it is mentioned that it is from the year 2014. 

It lacks sufficient theoretical support to directly support the relationships, which is why it is suggested to increase previous studies that support the seven hypotheses of the study.

 It is suggested to add a table with the hypotheses, which were accepted and rejected and reason for statistical agreement.

Author Response

Thank you very much for your constructive feedback.

We think the general topic of study concepts is valid across time, and hence, the data is very valuable and still interesting based on the large sample and study setting across four hospitals.

The findings are very clear since all hypotheses are supported across all hospitals, as such, we think the suggested table would be redundant since the presentation of results is already so clear and consistent.

We have tried to add some more theoretical support for the hypothesis, but the challenge is the novelty of the study which makes it more difficult to find relevant and good literature. Please also note the reference list is also already substantial with approximately 75 references.

Reviewer 3 Report

The issue taken up by the authors is topical and worth recognizing, particularly in terms of the negative relationship between job performance and job commitment and uncertainty among hospital employees. The article has some cognitive and applied value. However, the authors undertook empirical research in fairly recognized areas, as a result, the findings mainly confirm what has already been recognized. This is evident from the authors' discussion. It would be useful to describe and emphasize more extensively what is new, not yet recognized, and explain to what extent the conducted research fills the diagnosed research gap. It would be worth emphasizing more strongly the specifics of the conditions arising from the functioning of hospitals and the people working in them. The conclusions of the research are quite modest, especially the postulatory ones. It would be valuable to comment more extensively on the results of the study regarding the negative relationship between job performance and job commitment and uncertainty among hospital employees. It would be worthwhile to subject these results to a broader discussion. It is imperative to point out the limitations that the research findings have.

Author Response

Thank you for your kind comments and feedback. We are now less modest and have added more text and elaborated more in different parts of the paper. Please see changes in the uploaded new version of the paper.

Reviewer 4 Report

1. Introduction is too specific needs some elaboration

2. Theoretical reasoning is provided but lot of theories are mentioned authors need to explain whether they used one single theory or different theories and bridged?

3. Authors provided separate results for Hospital1, Hospital 2, hospital 3 and hospital 4, I think all together in one model is much better otherwise it would be comparative study then change title because you need to develop hypotheses accordingly. 

4.  Add policy implications 

5. Add Limitations and future directions 

Author Response

  1. Introduction is too specific needs some elaboration

Response:

Thank you for the review and comments on this manuscript.

The introduction is extended – see changes in the revised manuscript.

  1. Theoretical reasoning is provided but lot of theories are mentioned authors need to explain whether they used one single theory or different theories and bridged?

Response:

We developed a new model building of different theories and previous research. This is relatively common in SEM studies.

  1. Authors provided separate results for Hospital1, Hospital 2, hospital 3 and hospital 4, I think all together in one model is much better otherwise it would be comparative study then change title because you need to develop hypotheses accordingly. 

Response:

The aim is to test the validity of the model in sub-samples. Ref. this text in part 6:

One strength of the study relates to the testing of our theoretical model across four hospitals that are governed by the same laws and regulations but are managed by different line managers. Conclusively, the current study contributes to understanding the nomological network related to uncertainty and allows for the generalizability of the theoretical model within a Norwegian and possibly Scandinavian context.å

  1. Add policy implications 

Response: See changes in the discussion.

  1. Add Limitations and future directions

Response: Added – see changes in manuscript (new part 5.1).

Reviewer 5 Report

Theoretical introduction should be extended to include:

- human capital and its components,

- basic methods of improving employees' competences.

In the methodological part, the research period is missing.

Author Response

Thank you for the constructive review and comments.

We have extended more theory in the introduction, theoretical introduction and in the discussion.

We mentioned future studies on uncertainty can include other topics, such as human capital (see changes in part 5.1.

Basic methods of improving employees' competences have been elaborated more in the discussion with a new large paragraph – see changes in the manuscript.

Actually, the research period was already specified in this section. See below.

  1. Materials and methods

3.1. Sample and data collection

We collected data from four public hospitals in a Norwegian regional health authority. The regional health authority has more than 20,000 employees and provides services to 1.1 million citizens. The study was conducted in October 2014 and data was collected via an internal web-application distributed to all health care employees of the health authority.